# Research on factors affecting the spread of dust pollution in conveyor belt workshop and research on wet dust reduction technology

**Deji Jing**[1,2,3], **Jichuang Ma**[1,2,3], **Zhe Dong**[1,2,3]*, **Luyue Bai**[1,2,3], **Qisheng Kan**[1,2,3]

**1** College of Safety Science and Engineering, Liaoning Technical University, Fuxin, China, **2** Research Institute of Safety Science and Engineering, Liaoning Technical University, Fuxin, China, **3** Thermodynamic Disasters and Control of Ministry of Education, Liaoning Technical University, Fuxin, China

* 2099318066@qq.com

## Abstract

At this stage, there are many dust-hazardous industries, and occupational pneumoconiosis has a high incidence for a long time. To solve the dust pollution problem in coal processing plant workshops, the dust particle field and liquid droplet particle field were numerically simulated using computational fluid dynamics (CFD), and the influences of the induced airflow and corridor wind speed on the internal airflow field of the workshop were investigated to derive the dust pollution mechanism in the coal plant workshop under the change in the wind flow field. In this study, it was shown that the wind flow rate in the coal processing plant workshop is mainly affected by the corridor wind speed, and the higher the corridor wind speed is, the higher the wind flow rate. The induced airflow mainly affected the direction of the wind flow field in the workshop. According to the conclusions obtained from the simulations, a spray dust reduction system was designed for the coal processing plant workshop and applied in the Huangyuchuan coal processing plant. On-site measurement revealed that the dust reduction effect inside the coal processing plant workshop is obvious, and the overall dust reduction efficiency in the workshop reaches more than 94%, which meets the requirements of environmentally sustainable development and clean production.

## 1. Introduction

In recent years, the dust pollution problem in coal preparation plants has become increasingly serious, and dust pollution in coal preparation plants is mainly caused by the large amount of dust generated during belt conveyor transport in the process of production operations. Dust is extremely harmful to the human body, and workers and staff who remain in an operating environment with a high concentration of coal dust for a long time are very prone to coal workers' pneumoconiosis (CWP) [1,2], which greatly affects their physical and mental health and, in serious cases, even endangers their lives, resulting in enormous economic losses [3]. As shown in Fig 1, pneumoconiosis accounts for more than 80% of the total new cases of occupational diseases in China, which is much higher than that of other occupational diseases. At the same time, with dust accumulation, there is a risk of explosion; when a high concentration of dust

(contact via Liu Hongwei,email:734093674@qq.com) for researchers who meet the criteria for access to confidential data.

**Funding:** The authors gratefully acknowledge the financial support from the Liaoning Provincial Natural Science Foundation (2020-MS-304); Liaoning provincial funding for scientific research projects (LJK0323). The funders had no role in study design, data collection and analysis, decision to publish, or preparation of the manuscript.

**Competing interests:** The authors have declared that no competing interests exist.

meets an open flame, violent combustion can occur, leading to dust explosion [4], which threatens life and health. Dust not only causes harm to personnel but also causes damage to equipment, and dust adheres to the equipment surface to accelerate equipment wear and tear, reducing its service life [5]. In the dust pollution process in coal preparation plants mainly originating from the operation system of the coal transfer process, a large amount of dust is produced, and in this situation, high dust levels can easily accumulate in enclosed coal preparation plant spaces. In summary, research on dust management in the belt conveyor transport process in coal processing plants is crucial.

At present, many scholars have extensively studied dust management. Chen et al. [6] verified the feasibility of the CFD technique for dust emission prediction, showing that this technique can be used to qualitatively assess the performance of conveyor chute design in terms of dust emission. Chen et al. [7] simulated the distribution characteristics of droplets with varying particle sizes under different induced airflow conditions and determined the influence of the induced airflow on dust spraying. Nie et al. [8] used the CFD-DPM airflow–dust coupling

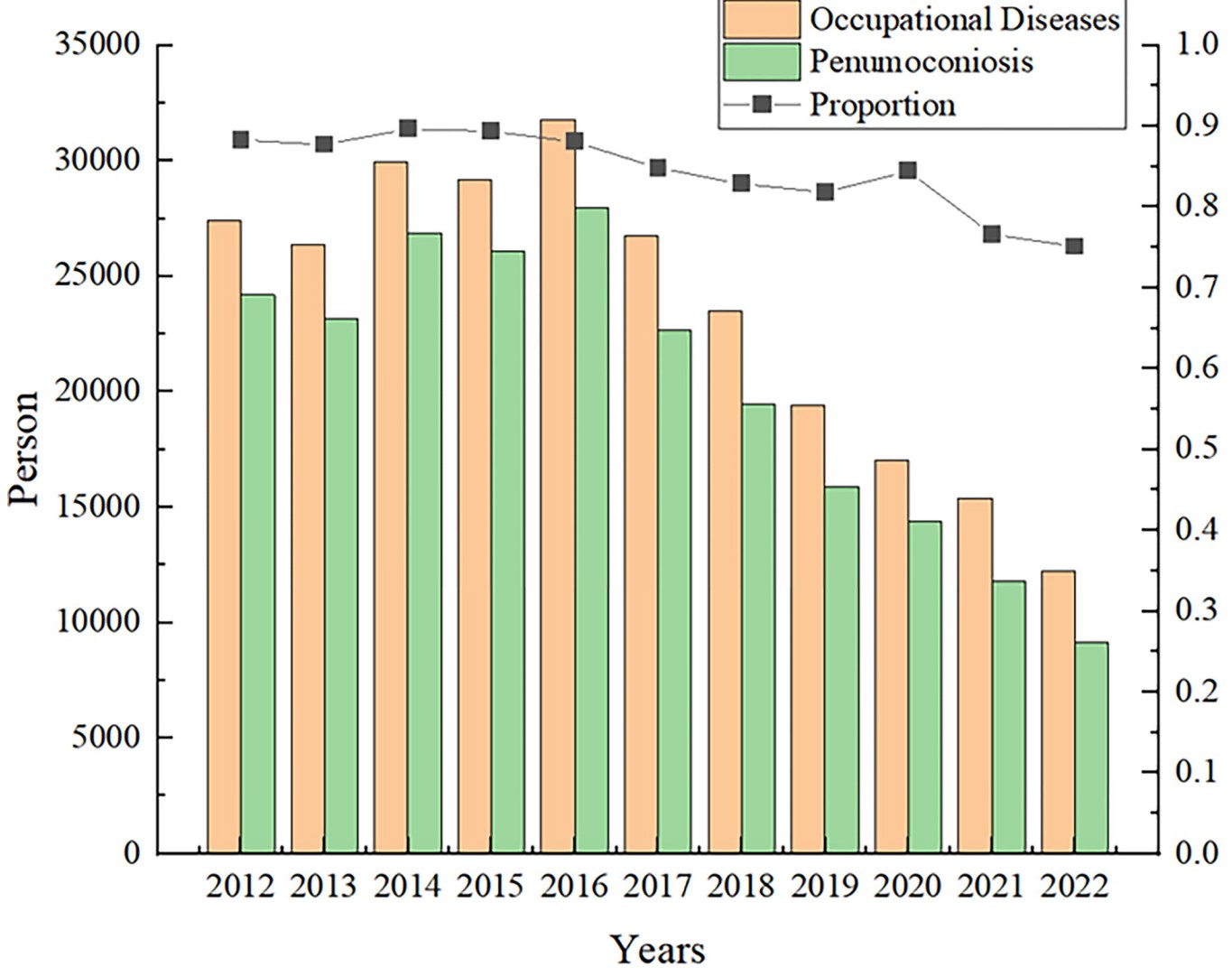

**Fig 1. Statistics of occupational disease cases in China over the past 10 years.**

method to study multisource diffusion and pollution of integrated coal mining faces and obtained the distribution trend of the dust particle size at different locations. Chen et al. [9] established a similar model for a conveyor belt conveyor roadway and found that the average wind speed in the roadway and the conveyor belt running speed were the two main factors affecting the dust concentration distribution. Bagnold [10] obtained a model of the relationship between the particle starting wind speed and particle size by mathematically calculating the torque balance generated when fluid flows between particles. Min et al. [11] used a gas–solid coupled dispersed multiphase flow model and field measurement data in simulation software (Fluent) to obtain the transport trend of respiratory dust. Wei et al. [12] studied the influence of different factors on the dust pollution conditions in the transport lane through similar experiments. Zhu et al. [13] investigated the dust distribution and transport trend under different boundary conditions through numerical simulations and found that the higher the conveyor belt running speed was, the higher the dust concentration. Cai P. [14] analyzed the coal dust particle pollutant diffusion situation under different airflow parameters based on numerical simulations with the CFD-DPM model, and the results showed that dust particles were more highly concentrated than other particles. The results indicated that the dust concentration is relatively low, and with increasing airflow, the phenomenon of secondary dust lifting could occur.

Zhang et al. [15] simulated the coal dust pollution behavior under the effect of system ventilation airflow (SVA) only and the addition of TACC perturbation and verified the simulation results by comparing them with field measurement data. The results showed that TACC perturbation not only enhances the turbulence intensity in the flow field but also induces a positive deviation in the airflow. Torno J A F I [16] analyzed the effect of dust emission on the loss of treatment products and maintenance of equipment and facilities. Du et al. [17] used the computational fluid dynamics (CFD) technique to simulate the airflow and dust movement across a mega-sized integrated mining face and proposed corresponding management measures. Fang et al. [18] found that the main dust trapping stages of fine water mist with different particle sizes differ in spraying dust reduction experiments. Zhang et al. [19] explored the effect of multistage external spraying on atomization and dust reduction in a comprehensive mining face based on computational fluid dynamics (CFD) theory, and the system could effectively reduce the dust concentration at the working face. Jing et al. [20] established a numerical model of the airflow–dust distribution through simulations to obtain the dust distribution in a coal processing plant, and they designed corresponding dust reduction schemes. Cao et al. [21] developed dust reduction equipment for the roadways of underground coal mines based on the principle of micron-level water mist dust reduction technology and found that this technology can yield a better dust reduction effect than ordinary spray water curtains.

Although many scholars have investigated dust transport and management at different locations in mines, parallel belt conveyor operation systems have not been analyzed. Parallel belt conveyor systems are used in most coal processing plants, so it is particularly important to study dust pollution originating from parallel belt conveyor operation systems. Therefore, in this paper, the factors affecting the wind flow field of the parallel belt conveyor operation system are analyzed, the wind flow and dust transport behavior of the parallel belt conveyor operation system is determined, and a theoretical basis is provided for dust management in coal processing plants. A spraying dust reduction device is designed for a parallel belt conveyor operating system in a coal processing plant, and the reliability of the numerical simulation results and the effectiveness of the designed dust reduction system are verified through on-site application, which lays a foundation for further development of a targeted dust reduction system.

## 2. Mathematical model

A model is developed considering airflow in the coal preparation plant through the connecting conveyor belt corridor to the workshop, airflow associated with workshop production processes, and airflow through the access stairway corridor door, lifting mouth and doors and windows to the outside. At the same time, material falling from the upper floor can partly induce the airflow from the first floor to the ground floor, which is mixed with the ambient airflow to form the background airflow field. Accordingly, based on the principle and discrete element method (DEM) of computational fluid dynamics (CFD), steady-state calculation of the background wind flow and the particle tracking method, the turbulence and particle motion module in COMSOL software is used for modeling and simulation purposes.

According to the theory of gas–solid two-phase flow and Newton's second law for systematic analysis of dust, dust particle dynamic movement in coal preparation plants can be expressed as follows:

$$m_p \frac{du_p}{dt} = F_g + F_f + F_d + F_x \tag{1}$$

where $m_p$ is the quality of dust, mg; $u_p$ is the velocity of dust, m/s; $F_d$ is the trailing force acting on dust, N; $t$ is the time, s; $F_g$ is the gravitational force acting on dust, N; $F_f$ is the buoyancy of dust in air, N; and $F_x$ denotes other forces, N.

Based on the Navier-Stokes gas control equation, the gas flow within this study is a turbulent flow with large Reynolds number, which is a complex vortex flow process, and it is a better choice to use the k-ε model to calculate the transient change of turbulent flow [22].

The momentum equation is:

$$\rho(u \cdot \nabla)u = \nabla \cdot [-pl + (\mu + \mu_T)(\nabla_u + (\nabla_u)^T)] + F = 0 \tag{2}$$

The turbulent kinetic energy ($k$) can be obtained as:

$$\rho(u \cdot \nabla)k = \nabla \cdot \left[ \left( \mu + \frac{\mu_T}{\sigma_k} \right) \nabla\kappa \right] + p_k - \rho\varepsilon \tag{3}$$

The turbulent energy dissipation rate ($\varepsilon$) can be expressed as:

$$\rho(u \cdot \nabla)\varepsilon = \nabla \cdot \left[ \left( \mu + \frac{\mu_T}{\sigma_\varepsilon} \right) \nabla\varepsilon \right] + c_{\varepsilon 1} \frac{\varepsilon}{k} p_K - c_{\varepsilon 2}\rho \frac{\varepsilon^2}{k}, \varepsilon = \text{ep}$$

$$\mu_T = \rho c_\mu \frac{k^2}{\varepsilon} \tag{4}$$

$$p_k = \mu_T[\nabla u : (\nabla u + (\nabla u)^T)]$$

where $\rho$ is the gas density, kg/m³; $k$ is the turbulent kinetic energy, m²/s²; $-\rho l$ is the Reynolds stress, $u$ is the wind velocity, m/s; F is the other external forces, N; $\mu$ is the Laminar viscosity coefficient, Pa·s; $\varepsilon$ is the turbulent kinetic energy dissipation rate, m²/s³; $P_k$ is the turbulent kinetic energy generation term due to the mean velocity gradient; $\mu_T$ is the turbulence viscosity coefficient, Pa·s; $C_\mu$, $C_{\varepsilon 1}$, and $C_{\varepsilon 2}$ are empirical constants, with $C_\mu = 0.09$, $C_{\varepsilon 1} = 1.44$, and $C_{\varepsilon 2} = 1.92$; $\sigma_k$ is the turbulent Prandtl number of $k$, with $\sigma_k = 1.0$; and $\sigma_\varepsilon$ is the Prandtl coefficient of $\varepsilon$, with $\sigma_\varepsilon = 1.3$.

The droplet model mainly depends on the nozzle type, which is one of the main factors affecting the effectiveness of spray dust reduction. According to the atomization dust reduction principle, the common nozzle types are pressure cyclone nozzles and gas–liquid two-phase nozzles. The gas–liquid two-phase nozzle produces a large proportion of small droplet

particles, the water consumption remains constant when the fog concentration is exponentially increased, and the wetting effect on coal dust is better than that of other nozzles. Therefore, in this paper, the gas–liquid two-phase nozzle is selected, and in numerical simulation of droplets, the Taylor analog breakdown (TAB) model [23] is used. According to this model, the droplet particle force equation can be expressed as:

$$\frac{d^2 y}{dt^2} = F_a - F_\sigma - F_\mu \tag{5}$$

$$F_a = \frac{C_F \rho |u - u_d|^2}{C_b \rho_d r^2} \tag{6}$$

$$F_\sigma = \frac{C_K \sigma}{\rho_d r^3} y \tag{7}$$

$$F_\mu = \frac{C_d \mu_d}{\rho_d r^2} \frac{d_y}{d_t} \tag{8}$$

where $F_a$ is the aerodynamic force, N; $F_\sigma$ is the surface tension, N; $F_\mu$ is the viscous force, N; $\mu_d$ is the kinetic viscosity of the droplet, kg/m-s; $\rho$ is the gas density, kg/m$^3$; $\rho_d$ is the air-liquid droplet density, kg/m$^3$; $u_d$ is the droplet velocity, m/s; r is the droplet radius, m; and $C_F$, $C_b$, $C_K$ and $C_d$ are constants, with $C_F$ = 1/3, $C_b$ = 1/2, $C_K$ = 8 and $C_d$ = 5. For y = 1, i.e., the displacement x is half of the initial droplet radius r (x = 0.5r), and the droplet is considered to have broken.

## 3. Construction of the computing domain and parameter settings

### 3.1 Geometric model establishment

The established geometric model consists of two conveyor belts, five discharge ports, conveyor belt corridor, conveyor belt sealing groove, machine tail, wall, dust curtain at the connection of the conveyor belt corridor, and a large structure on the left side of the machine. The 3001 and 3002 conveyor belt conveyors share the conveyor belt corridor, and the corridor and conveyor belts within are set at a certain angle to the horizontal plane, namely, approximately 20°. The length of the 3001 conveyor belt part of the conveyor belt corridor is approximately 33 m, and the length of the 3002 part is approximately 22 m. The conveyor belt corridor initially encompasses a horizontal section of approximately 5 m, and the length of the angled section is set to approximately 5 m. The height of the workshop is 3.8 m, and the upper edge of the conveyor belt sealing groove is approximately 2 m from the ground. Other geometric modeling details are set according to the actual site measurements. The spatial position of the main equipment and structures is shown in Fig 2. The model inlet is the drop inlet, as well as the left side of the corridor into the air, the model outlet is the workshop on the right side of the workshop in and out of the location, the model is surrounded by walls inside the model without walls to divide the model.

### 3.2 Grid generation and independence test

In computational meshing, both the computational accuracy and computational efficiency must be considered. The denser the grid is, the higher the computational accuracy, and at the same time, the higher the computational load. Therefore, three groups of grid parameters (A, B, and C) can be obtained by changing the grid parameters: 953617 grids in Group A, 2516955 grids in Group B, and 10648907 grids in Group C. The cell mass values of the three grid groups

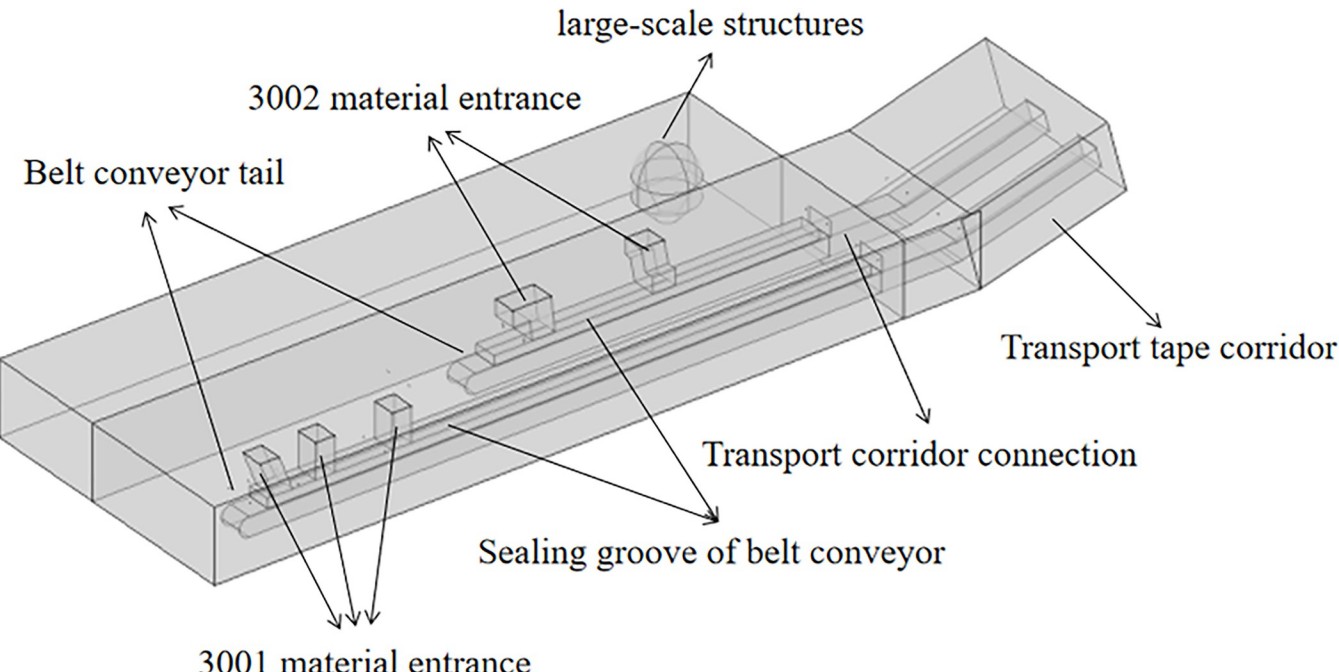

**Fig 2. Geometric numerical simulation model for dust pollution in a workshop.**

all vary between 0.4 and 1, and the grid quality is satisfactory, which meets the computational requirements. As shown in Fig 3, The airflow velocity of grid B does not differ much from that of grid C and is less affected by the grid. Hence, grid B meets the calculation requirements, and the increase in the number of grids does not significantly improve the calculation accuracy. Therefore, grid B can be selected for the calculations.

## 3.3 Boundary conditions and parameter settings

The conveyor belt runs in the positive direction along the x-axis, and the model wall conditions are defined as fixed with no rebound of dust particles. The set numerical simulation boundary conditions are listed in Table 1.

## 3.4 Reliability verification analysis

To obtain correct conclusions from the numerical simulations, it is important to ensure that the model accurately reflects reality when performing calculations and analyses. Therefore, the validity of the model must be evaluated. In this study, the model is validated based on the wind flow conditions, and the model validity is verified by comparing the numerical simulation and model test results. When the flow field in the workspace remains relatively stable, five measurement points can be selected for validation. Take the left side of the coal processing workshop as the reference 5 meters to arrange the measurement point 1, 15 meters for the measurement point 2, 20 meters for the measurement point 3, 30 meters for the measurement point 4, 40 meters for the measurement point 5, the use of hand-held anemometer. Triplicate measurements are conducted at each measurement point, and the average value is compared with the simulated wind flow velocity data of the corresponding measurement point. The results are shown in Fig 4. The comparison study reveals that the maximum error at the measurement points between the wind flow velocity data and the simulation results is only 8.34%,

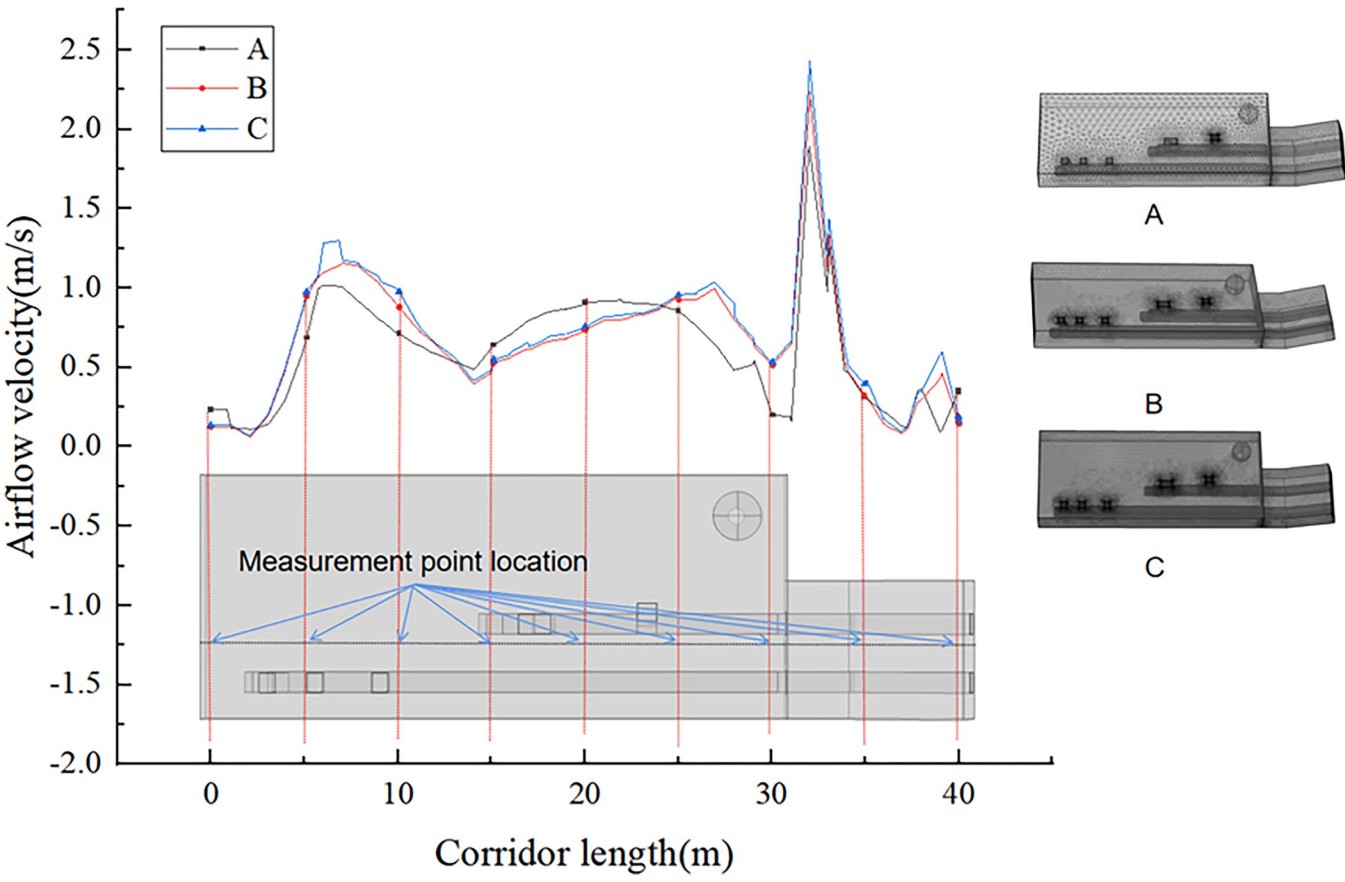

**Fig 3. Grid division.**

which indicates that the numerical model is consistent with the actual situation. Therefore, it can be concluded that the numerical model is accurate.

## 4. Numerical simulation and airflow–dust–droplet transport trend result analysis

### 4.1 Analysis of the wind flow field influencing factors

To obtain the factors affecting the wind flow transport trend in the workshop of the coal processing plant during operation, the influence mechanism was analyzed. By varying the

**Table 1. Boundary condition setting.**

| Boundary conditions | Set values |
|---|---|
| Drop induced air velocity/m·s$^{-1}$ | 0.58 |
| Roughness coefficient | 0.26 |
| Coal flow height/mm | 100 |
| conveyor belt running speed/m·s$^{-1}$ | 3.27 |
| Density of coal dust/kg·m$^{-3}$ | 1.63 |
| Air density/kg·m$^{-3}$ | 1.27 |
| Dynamic viscosity/Pa·s | $1.814 \times 10^{-5}$ |
| Wall mounting | No-slip |
| Temp/K | 293.15 |

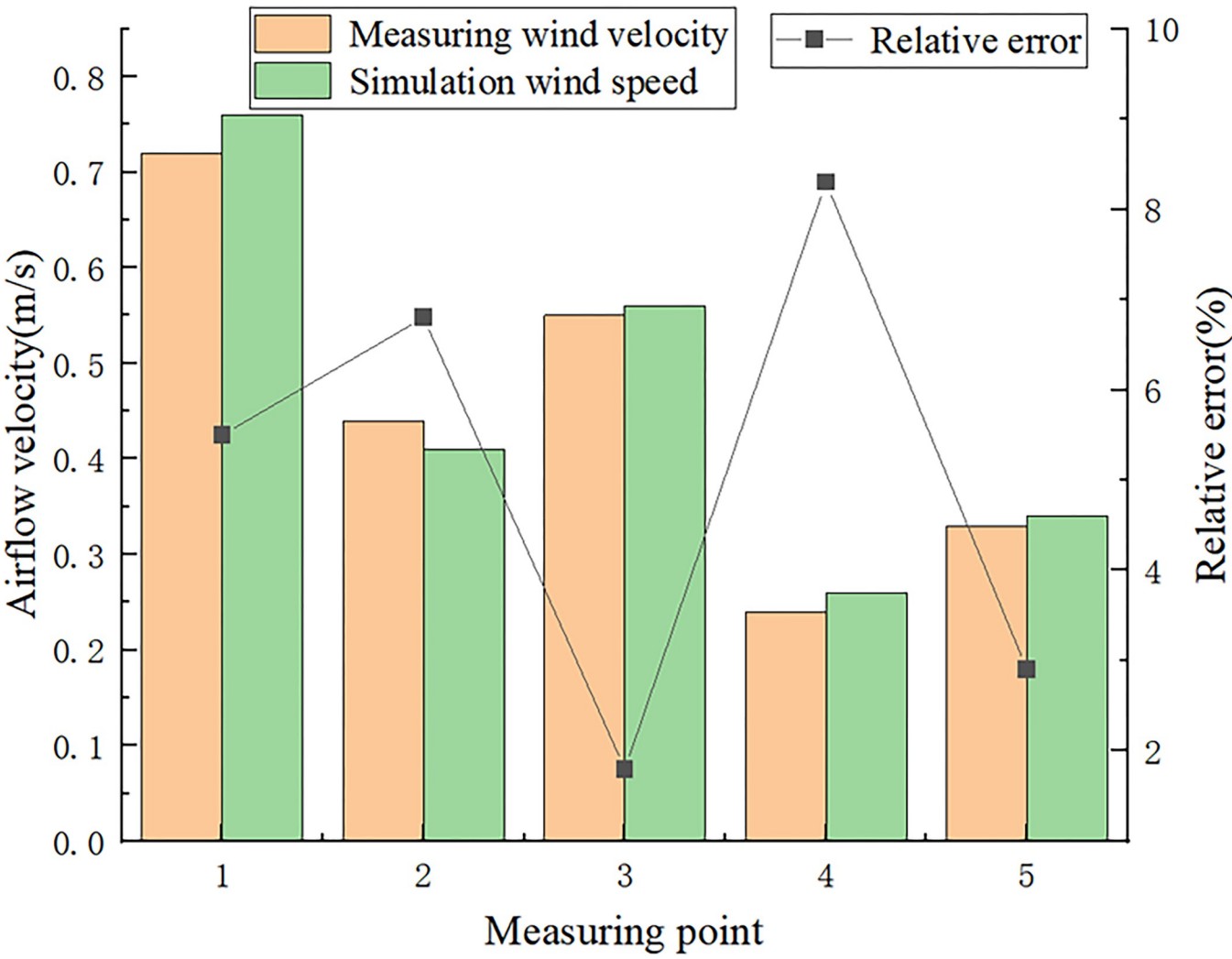

**Fig 4. Model validation.**

magnitude of the induced airflow and the corridor wind speed, several numerical simulations were conducted to obtain a flow diagram of the wind flow field in the workshop under different parameter conditions, and the simulation results were analyzed to determine the influencing mechanism of different factors on the wind flow transport trend in the coal processing plant during belt conveyor operation.

Due to the principle of incompressible flow, when the airflow enters the workshop from the conveyor belt corridor, the airflow in the conveyor belt corridor near the inner side of the workshop to the end of the conveyor belt machine at the conveyor belt surface and ground is accelerated, while the airflow speed in other areas is low. Moreover, the inward airflow speed from the opening of the conveyor belt corridor gradually decreases. The airflow at the end of the conveyor belt machine is slightly higher than that at other locations due to the influence of the return of the secondary airflow. The wind flow induced by the two transport conveyor belts during operation extends from the conveyor belt expansion to the middle of the two conveyor belts above and the right side of the wall to the collision point, thus generating a vortex accumulation area outside of the transport conveyor belts. Fig 5 shows that when the wind

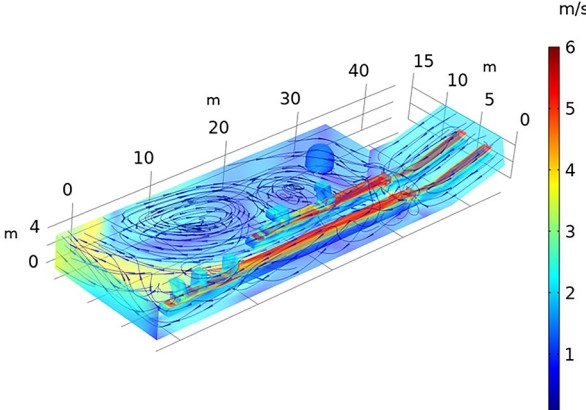

(a) Induced airflow 0.36m/s, corridor wind speed 0.3m/s

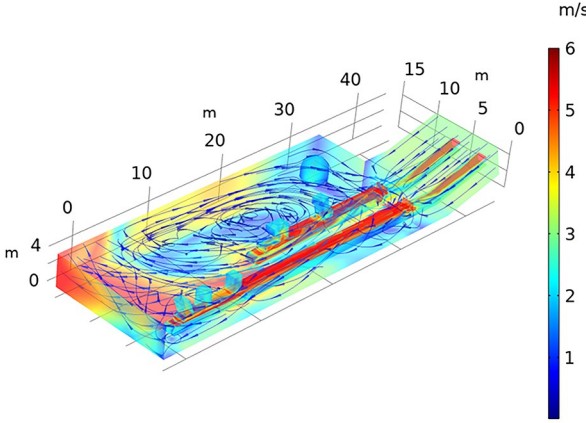

(b) Induced airflow 0.36m/s, corridor wind speed 0.5m/s

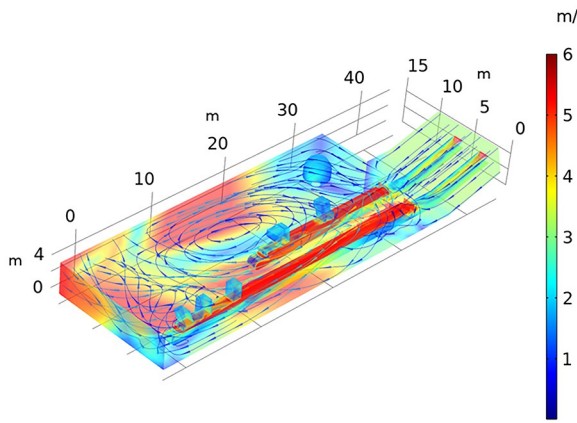

(c) Induced airflow 0.36m/s, corridor wind speed 0.8m/s

**Fig 5. Effect of the corridor wind speed on the wind flow field.** (a)Induced airflow 0.36m/s, corridor wind speed 0.3m/s. (b)Induced airflow 0.36m/s, corridor wind speed 0.5m/s. (c)Induced airflow 0.36m/s, corridor wind speed 0.8m/s.

speed in the workshop corridor increases, the vortex accumulation area of the airflow field in the workshop tends to remain stable. The air velocity in the workshop increases, and each position in the workshop is affected similarly. The conveyor belt conveyor tail is affected the most, followed by the conveyor belt outside. With increasing wind speed, the maximum flow rate in the workshop increases from 4 to 6 m/s. Due to the influence of the large structures in the lower part on the wind flow area, the wind speed does not notably increase.

When the conveyor belt machine in the coal processing plant workshop is operated, the high-speed conveyor belt drives the surrounding air flow, and this type of airflow is denoted as induced airflow. The induced airflow mainly disturbs the corridor wind flow to alter the stability of the flow field in the workshop and the airflow direction, as shown in Fig 6. With increasing induced airflow, the airflow velocity above the conveyor belt decreases, the overall wind speed change in the workshop wind flow field does not significantly increase, and the overall wind speed remains below 3 m/s. With increasing induced airflow in the workshop, the magnitude of the vortex accumulation area of the airflow field in the workshop is reduced until it is destroyed to form a chaotic flow field. With increasing induced airflow, the airflow velocity of the airflow field on the outside of the conveyor belt machine does not obviously change, but careful observation reveals that the airflow velocity slightly decreases because the induced airflow cancels out the corridor airflow.

The above phenomena are analyzed to determine the influence mechanism of conveyor belt machine operation, corridor wind speed changes and induced airflow on the wind flow field in the coal plant workshop. In the wind flow field of the workshop, the flow rate is mainly affected by the corridor wind speed. The higher the corridor wind speed is, the higher the wind flow rate. The induced airflow mainly affects the direction of the wind flow field in the workshop. The induced airflow exerts a counteracting effect on the corridor airflow, and the greater the induced airflow, the less prone the area is to vortex accumulation. Notably, the magnitude of the induced airflow and corridor wind speed can be adjusted to control the overall flow field in the coal plant.

## 4.2 Dust particle field analysis

The influence of the airflow field on dust diffusion was analyzed under different conditions. To this end, dust particle diffusion under different flow fields was obtained by varying the corridor wind velocity and the magnitude of the induced airflow to determine the influence mechanism of different factors on the dust particle transport pattern in the coal processing plant workshop during conveyor belt machine operation. It was found that the distance and kinetic energy of dust particles transported in the different airflow fields vary, but Figs 7 and 8 show that regardless of increasing the corridor wind speed or the magnitude of the induced airflow, the direction of dust particle transport changes, and dust particles increasingly accumulate at the end of the belt conveyor in the coal processing plant.

Fig 7 reveals that the transport distance of dust particles increases with increasing corridor wind speed. When the wind speed in the corridor reaches 0.5 m/s, dust particle pollution at the tail of the conveyor belt machine in the coal preparation plant is the highest. When the corridor wind speed is further increased to 0.8 m/s, dust particle accumulation at the end of the conveyor belt machine is reduced, but particle diffusion occurs at the middle of the workshop. Dust particles are mainly concentrated at the end of the conveyor belt machine, on both sides of the transport conveyor belt and near the large structures.

The comparison of Figs 7 and 8 shows that the locations of dust production points in the coal processing plant are basically the same, dust particles mainly occur around the two conveyor belts, and dust particles exhibit higher kinetic energy when the corridor wind speed

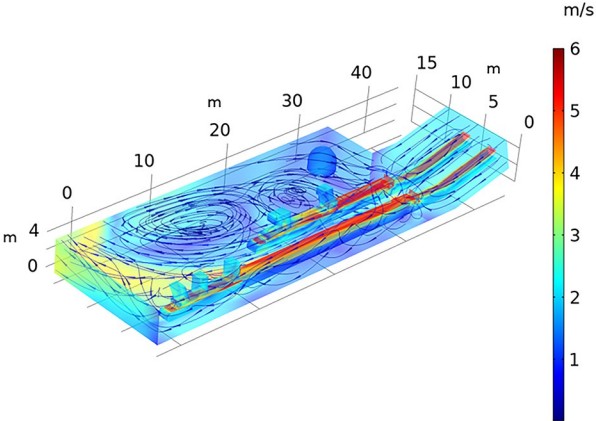

(a) Induced airflow 0.36m/s, corridor wind speed 0.3m/s

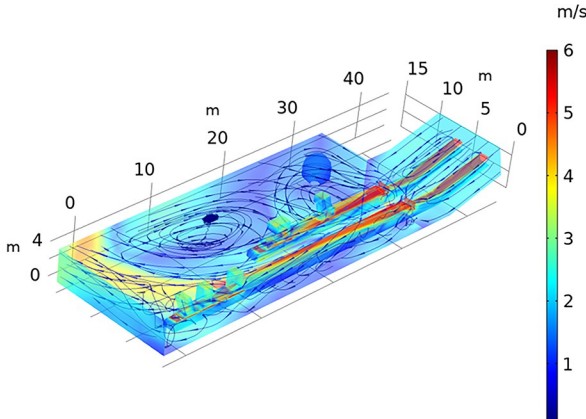

(b) Induced airflow 0.56m/s, corridor wind speed 0.3m/s

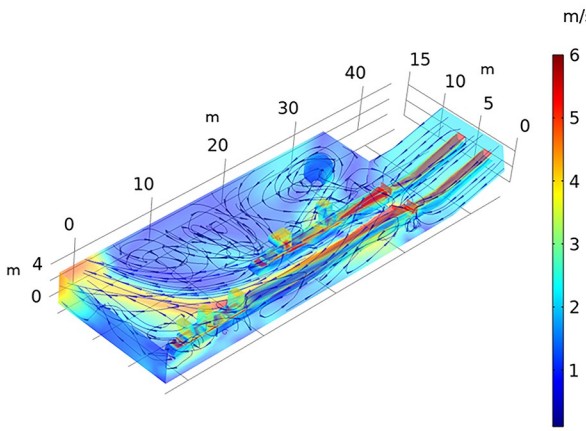

(c) Induced airflow 0.76m/s, corridor wind speed 0.3m/s

**Fig 6. Effect of the induced airflow on the wind flow field.** (a)Induced airflow 0.36m/s, corridor wind speed 0.3m/s. (b)Induced airflow 0.56m/s, corridor wind speed 0.3m/s. (c)Induced airflow 0.76m/s, corridor wind speed 0.3m/s.

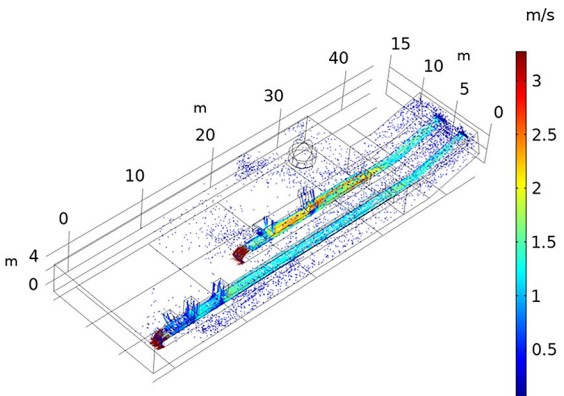

(a) Induced airflow 0.36m/s, corridor wind speed 0.3m/s

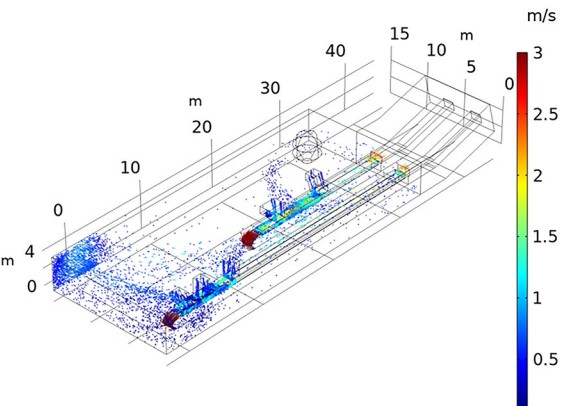

(b) Induced airflow 0.36m/s, corridor wind speed 0.5m/s

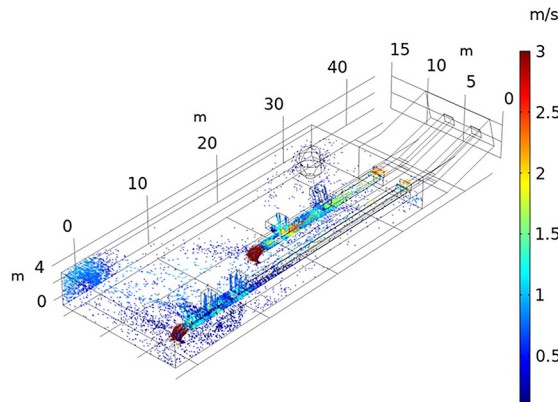

(c) Induced airflow 0.36m/s, corridor wind speed 0.8m/s

**Fig 7. Effect of the corridor wind speed on particle transport.** (a)Induced airflow 0.36m/s, corridor wind speed 0.3m/s. (b)Induced airflow 0.36m/s, corridor wind speed 0.5m/s. (c)Induced airflow 0.36m/s, corridor wind speed 0.8m/s.

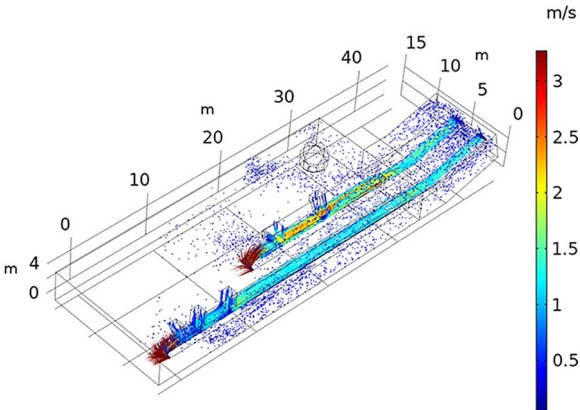

(a) Induced airflow 0.36m/s, corridor wind speed 0.3m/s

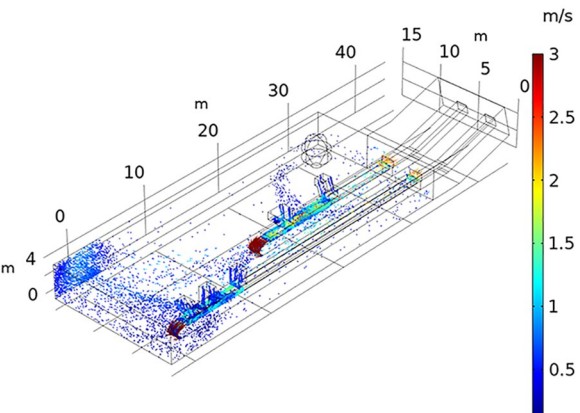

(b) Induced airflow 0.56m/s, corridor wind speed 0.3m/s

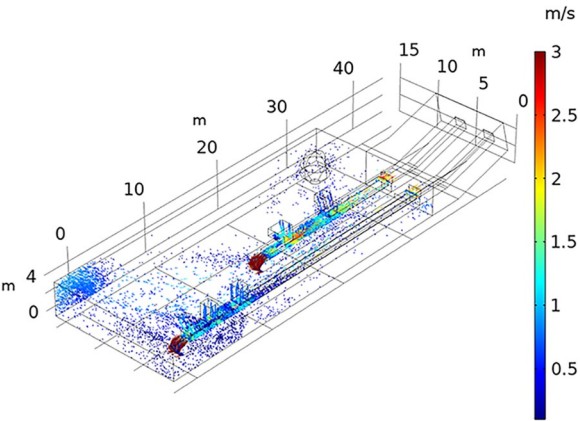

(c) Induced airflow 0.76m/s, corridor wind speed 0.3m/s

**Fig 8. Effect of the induced airflow on particle transport.** (a)Induced airflow 0.36m/s, corridor wind speed 0.3m/s. (b)Induced airflow 0.56m/s, corridor wind speed 0.3m/s. (c)Induced airflow 0.76m/s, corridor wind speed 0.3m/s.

increases relative to the induced airflow, which suggests that dust particles are transported across a larger distance and are more difficult to capture. When the induced airflow or the corridor wind speed continues to increase, the coverage of dust particles in the workshop increases, the movement speed of dust particles increases, and the dust movement speed in the lower left corner of the workshop increases from 0.5 to 1 m/s. Notably, the movement of dust particles is jointly determined by the induced airflow and corridor wind speed. The comparison of Figs 7 and 8 indicates that the effects of the induced airflow and corridor wind speed on dust particle movement are similar, and the corridor wind speed largely determines the flow velocity in the workshop, but it is not the dominant factor of the transport distance of dust particles.

## 4.3 Analysis of the droplet diffusion trend

According to the simulation results of dust pollution in the coal preparation plant workshop under different conditions, the pollution sources of the coal preparation plant workshop are summarized and divided. Due to the many dust-producing points in the workshop and the complexity of dust particle transport, a composite spraying device is used for control of the dust source points. The workshop interior is divided for multilevel spray dust reduction device installation considering the areas with serious pollution, and spray devices are installed in the vicinity of the discharge port, the end of the conveyor belt machine, the conveyor belt corridor connection and other locations. According to the installation of spraying dust reduction devices, simulations are conducted.

Fig 9 shows the water mist particle field under the different wind flow fields. The water mist particle size is smaller than 50 μm, and 35-μm water mist particles account for more than 90% of this water mist particle size range. Thus, a high efficiency of dust capture is achieved. When the induced airflow and corridor wind speed are low, dust particles mainly float in the conveyor belt corridor. At this time, as shown in Fig 9A, the water mist particles form a water curtain in the conveyor belt corridor connection to prevent dust particles from greatly penetrating the conveyor belt corridor. Water mist particles are mainly concentrated in the 30-m conveyor belt corridor. Through dust and water mist particle field comparison, it was found that the coverage of water mist particles is much higher than that of dust particles.

As shown in Fig 9B, when the corridor wind speed was increased to 0.5 m/s, the induced airflow increased to 0.56 m/s, and dust particles originating from the conveyor belt corridor connection accumulated in the lower left corner of the workshop. At this time, the diffusion pattern of water mist particles also changed, water mist particles traveled to the bottom of the workshop due to the wind flow, and water mist floating clouds were formed at the bottom of the workshop. Compared with the initial coverage rate of water mist particles in the workshop, the coverage rate of water mist particles was greatly improved, and a vortex accumulation area was formed at 20 m in the workshop, which enhanced the entrainment of dust particles and improved the dust catching efficiency. Under the different wind flow conditions in the workshop, the movement range of water mist particles was basically similar to that of dust particles, and the coverage in the workshop was higher than that of dust particles, which shows that this dust reduction program favorably affects dust reduction.

## 5. Engineering application

According to the above numerical simulation results, the particle field of droplets produced by the spray device installed in the coal processing plant basically conforms with that of dust, which yields a suitable dust reduction effect. However, to illustrate whether the spraying device produces a reasonable dust reduction effect in the coal processing plant workshop, the

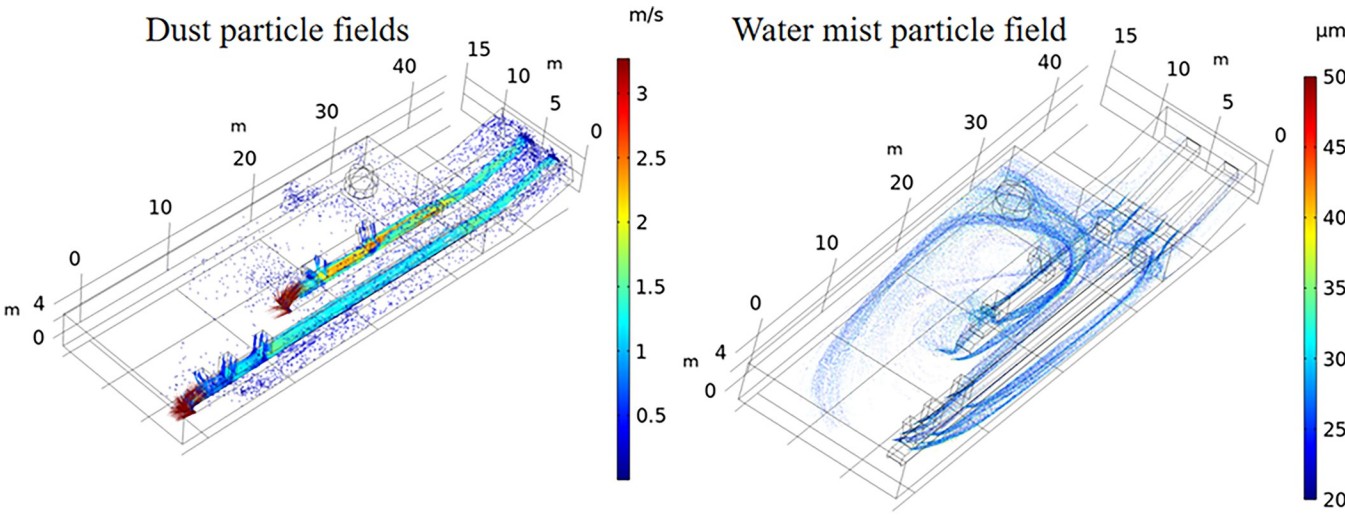

(a) Induced airflow of 0.36 m/s, corridor wind speed of 0.3 m/s

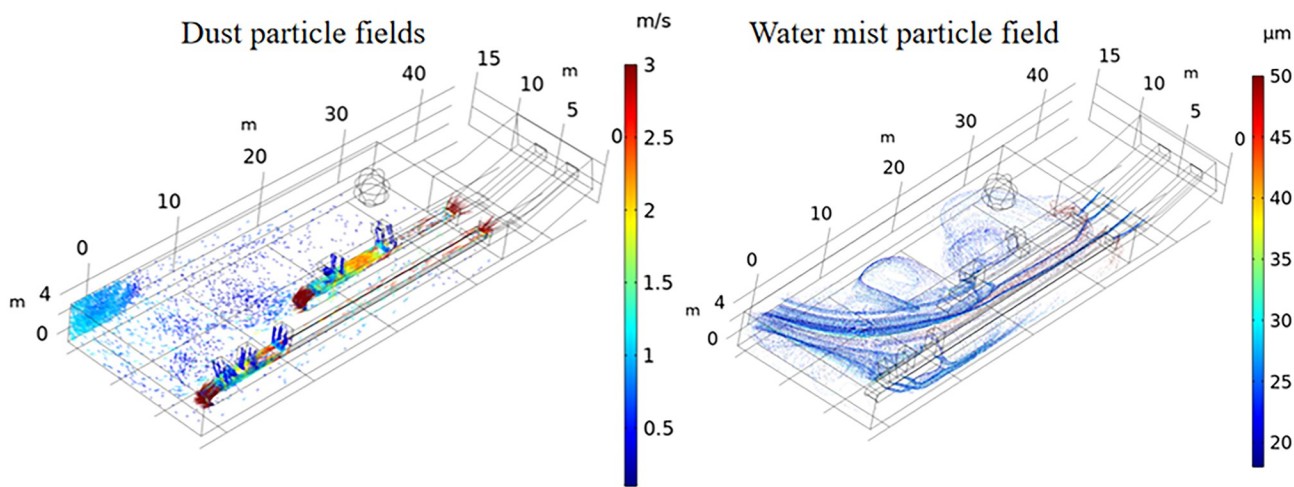

(b) Induced airflow of 0.56 m/s, corridor wind speed of 0.5 m/s

**Fig 9. Transport of water mist particles under different conditions.** (a) Induced airflow of 0.36 m/s, corridor wind speed of 0.3 m/s. (b) Induced airflow of 0.56 m/s, corridor wind speed of 0.5 m/s.

workshop of the Huangyu Chuan coal processing plant was selected for on-site application and data measurement. Dust pollution at the Huangyu Chuan coal processing plant is serious mainly due to the multiple dust sources coupled with diffusion, and during production, high airflow occurs in the conveyor belt corridor to the workshop. Falling material and the conveyor belt traction-induced airflow produce a turbulent wind flow field, and suspended respiratory coal dust with a small particle size can hardly escape from airflow entrainment due to turbulence in the area, resulting in accumulation.

The dust reduction system mainly selects pneumatic atomizing nozzle. As the water mist particle size of the pneumatic atomizing nozzle ranges from 8 to 35μm, it is more suitable for the capture of fine dust. In the installation of the spray dust reduction system in the area with serious dust pollution, nozzles are arranged mainly at three locations, namely, near the dust reduction area at the discharge port, the dust reduction area at the end of the conveyor belt machine, and the dust reduction area in the conveyor belt corridor. In the process of raw coal screening, a large amount of secondary coal dust is generated due to impact, friction and crushing, and lump coal transport induces a surrounding airflow under the effect of gravity as it is ejected onto the belt conveyor, forming a high-intensity impact-induced airflow at the conveyor belt surface. A positive-pressure impact-related airflow that can carry dust is produced in the sealing groove from the conveyor belt soft connection discharge point to the outside. Due to the high impact force, the positive-pressure airflow accelerates the coal dust jet through the relatively small opening of the leakage point into the environment. When the belt conveyor is operated, fine coal dust particles are deposited on the conveyor belt surface and in the machine tail section although it does not contain a coal-carrying belt. Fine coal dust particles are generated by the falling material and the tail centrifugal force of the production process, which adhere to the surface of the conveyor belt during belt conveyor motor operation. These particles can be detached from the conveyor belt due to vibration and gravity forces and can be entrained and transported by the combined airflow comprising traction-induced airflow and impact-induced airflow. When the belt conveyor motor is running, fine coal dust particles are separated from the belt by vibration and gravity and transported with the combined airflow of traction-induced airflow and impact-induced airflow. The conveyor belt corridor

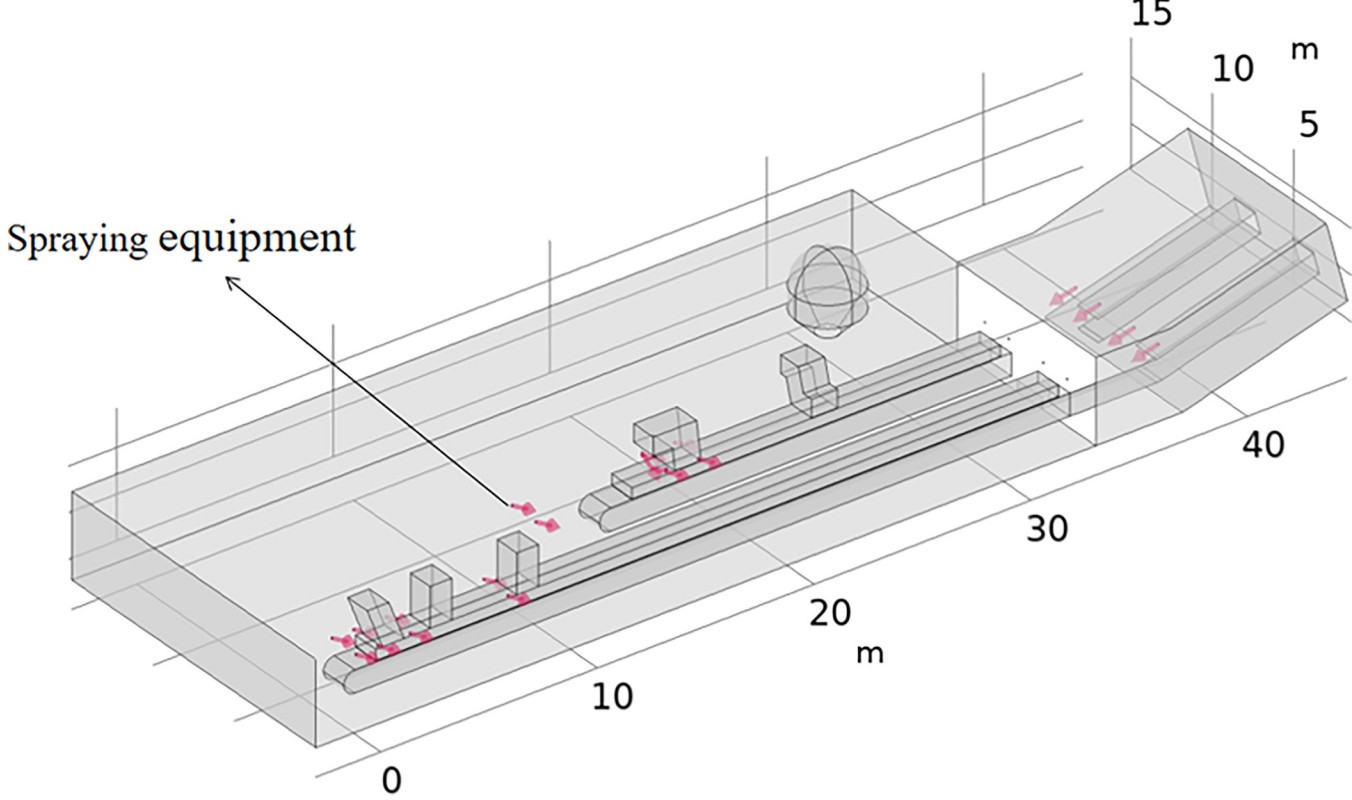

**Fig 10. Pneumatic atomizing nozzle location diagram.**

and workshop are separated by a dust curtain. After the start of production, the wind flow enters the workshop through the gap in the dust curtain, and the dust curtain is shifted toward the inner side of the sealing groove of the conveyor belt conveyor. Therefore, the ambient wind flow and the induced airflow at the border between the two parts jointly generate local turbulence, which causes dust gathering.

The specific arrangement is as follows: two rows of eight pneumatic fogging nozzles are arranged in parallel on both sides of the sealing groove of the belt conveyor, slightly higher than the top of the sealing groove, namely, at a distance of 2.3 m from the ground. They are arranged at an angle of 45° with the pipeline, with the direction of jetting following the coal flow direction of the belt conveyor. At the end of the belt conveyor, there are 6 pneumatic fogging nozzles located on the left, right and front sides of the tail, two on each side, which are installed based on the site conditions. On the inside of the conveyor belt corridor, there are 4 pneumatic fogging nozzles, 2 on each of the two conveyor belts. The arrangement position is approximately 1~1.5 m away from the conveyor belt, and the spraying direction is vertically downward and directly onto the conveyor belt to avoid large-scale wetting of the coal and conveyor belt. The total number of pneumatic atomizing nozzles is 18, the whole system consumes

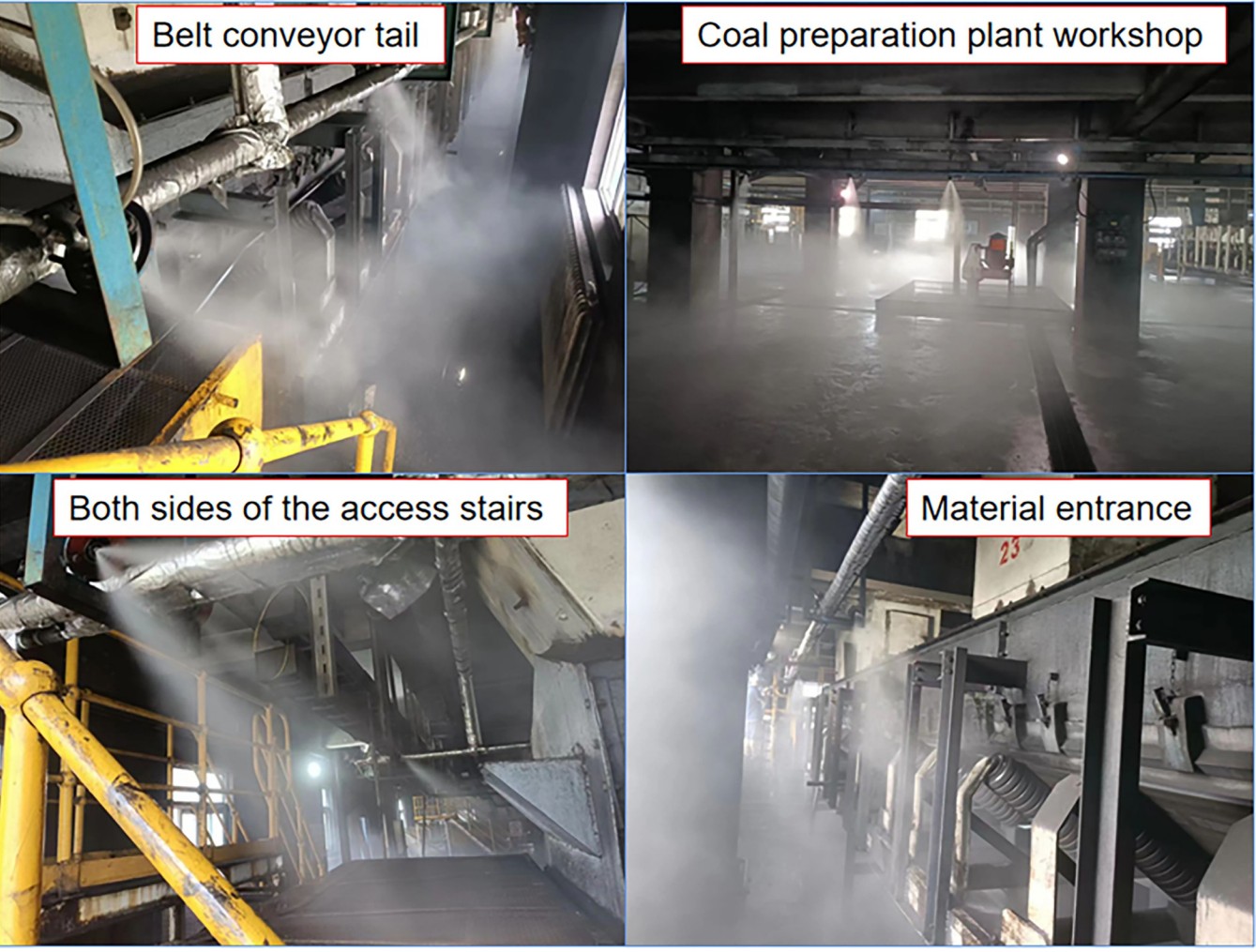

**Fig 11. Engineering application.**

approximately 90 m³/h of water, and the total water consumption is approximately 3.5 L/min. The location of the pneumatic atomizing nozzle is shown in Fig 10.

When the spray dust reduction device in the coal plant workshop is activated, the atomized spray fog curtain (along the edge of the machine tail) covers the machine tail on both sides of the spray. At the 3001 and 3002 conveyor belt machine tails, a high-speed aerosol curtain is formed due to gravity, the tail and outlet of the dust leakage point are fully surrounded by a curtain of fog droplets given a sufficient power supply, and based on the droplet speed characteristics, a high-efficiency and low-humidity coal dust capture effect is achieved. The jet spray covers the dust path at the tail, diminishing the total kinetic energy of the tail-flung dust and the induced airflow and pulling the stripped coal dust into the spray flow field under the negative pressure. When the belt conveyor is operated, the fog screen in the front and rear sections of the belt corridor is affected by the negative-pressure traction-induced wind flow and is sucked into the belt corridor from the workshop through the gap between the sealing groove and the belt corridor connection, covering the space near the top of the coal flow and forming relatively static movement conditions along the same direction as the coal flow direction. The coal dust on the surface of the belt is stripped by the wind flow, which produces a flying phenomenon in the coal flow transport process. The overall effect is shown in Fig 11, which reveals that a favorable water mist coverage is realized in different areas.

Measuring points are arranged according to the onsite equipment placement and the dust transport area, and a total of five measuring positions are established, as shown in Fig 12, namely, at the tail of the two conveyor belt machines, the connection of the conveyor belt

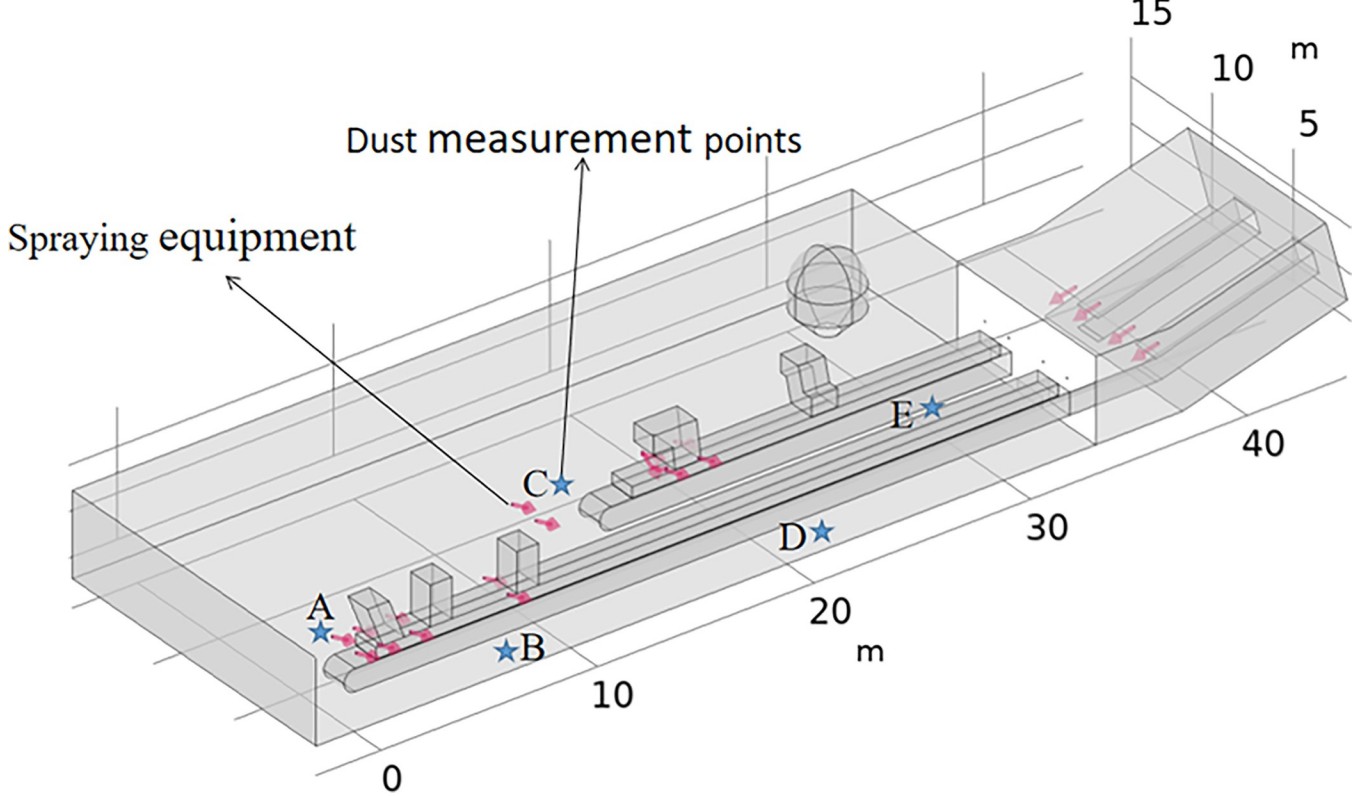

**Fig 12. Schematic layout of the measurement points.**

corridor, and the discharge opening of the conveyor belt machine. The height of the measurement points is 1.6 m from the ground, considering the workers' breathing zone. The most accurate dust sampling–drying–weighing method was used for dust concentration measurement to prevent the spray and moisture levels in the air from affecting the dust quality measurement results. To avoid the influence of random errors, each measurement point was sampled three times, and the average value was adopted as the final measurement result.

As shown in Fig 13, when the spraying dust reduction device is activated for some time, the dust removal efficiency in the area of point B is the highest, at 97.39%, and the dust reduction efficiency in the area of points A, C and D is above 95%. Only the area of point E exhibits the lowest dust reduction efficiency, at 94.47%. Under these conditions, the dust removal efficiency is significantly improved, and the dust pollution problem in the workshop of the Huangyu Chuan coal processing plant is effectively mitigated.

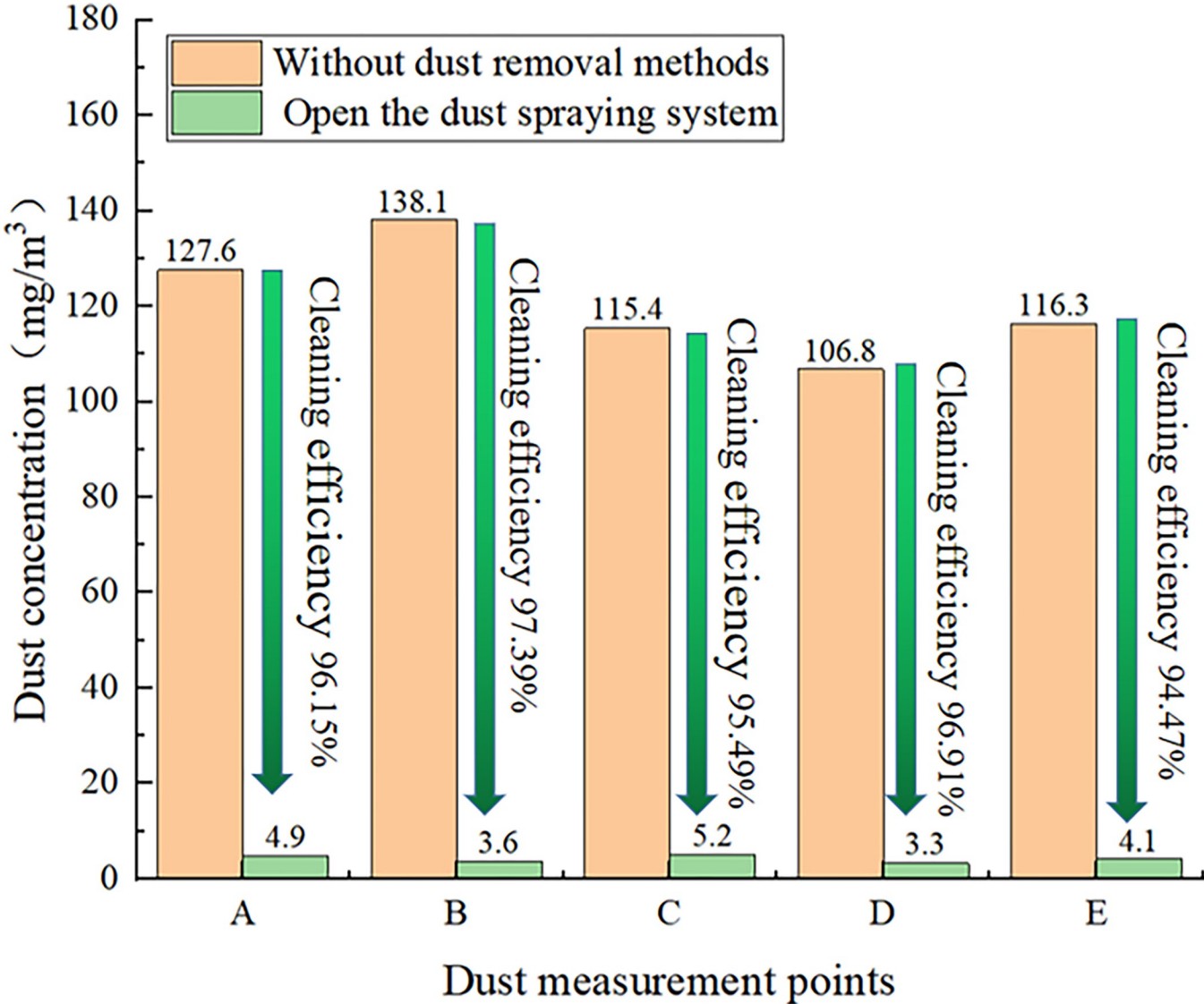

**Fig 13. Histogram of the dust reduction efficiency.**

## 6. Conclusions

In this study, a numerical simulation model of dust pollution in the workshop of a coal processing plant was established to analyze the effects of the induced airflow and corridor wind speed on dust pollution in the coal processing plant workshop. Based on the simulation results, a dust reduction system suitable for coal processing plant workshops was designed and developed. It was applied in the workshop of the Huangyuchuan coal processing plant, and the dust reduction effect was measured on site. Based on the above research, the following conclusions can be drawn:

1. At the coal processing plant, the induced airflow and the corridor wind speed jointly determine the wind flow field. The corridor wind speed is the decisive factor of the wind flow field affecting the flow rate. The higher the velocity of the wind flow field is, the larger the transport distance of dust particles. The induced airflow exerts a counteracting effect on the corridor airflow, and the higher the induced airflow, the less likely the vortex accumulation area is, reducing the dust gathering situation.

2. According to the numerically simulated distribution of dust particles, a targeted dust reduction device arrangement scheme is proposed. Serious dust pollution mainly includes three locations, namely, near the dust reduction area of the discharge port, the dust reduction area of the tape machine tail, and the dust reduction area of the tape corridor. According to the different positions of the targeted arrangement of spraying dust reduction devices, determined the coal plant workshop fog curtain dust control system. When the corridor wind speed and magnitude of the induced airflow are optimized, the transport trends of water mist and dust particles basically match. Hence, this dust reduction program can provide a satisfactory dust reduction effect.

3. Through the analysis of the numerical simulation results, the changes in the dust concentration before and after application of the spraying dust reduction system in the raw coal workshop of the Huangyuchuan coal processing plant were measured. The dust concentration at five measurement points was significantly reduced. The dust removal rate in each part reached more than 94%, in which the highest dust reduction efficiency reached 97.39%, and the dust concentration in the workshop was effectively reduced. This system can meet the requirements of environmentally sustainable development and clean production at coal preparation plants.

## Author Contributions

**Conceptualization:** Deji Jing.

**Data curation:** Jichuang Ma.

**Formal analysis:** Zhe Dong.

**Methodology:** Luyue Bai.

**Validation:** Qisheng Kan.

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
