## [Decision Letter · Decision Letter 0]

19 Dec 2023

PONE-D-23-38475Research on factors affecting the spread of dust pollution in conveyor belt workshop and research on wet dust reduction technologyPLOS ONE

Dear Dr. dong,

Thank you for submitting your manuscript to PLOS ONE. After careful consideration, we feel that it has merit but does not fully meet PLOS ONE’s publication criteria as it currently stands. Therefore, we invite you to submit a revised version of the manuscript that addresses the points raised during the review process.

ACADEMIC EDITOR: Major revision.

We look forward to receiving your revised manuscript.

Kind regards,

Worradorn Phairuang, Ph.D.

Academic Editor

PLOS ONE

Journal Requirements:

3. Thank you for stating the following financial disclosure: "The authors gratefully acknowledge the financial support from the Liaoning Provincial Natural Science Foundation (2020-MS-304);Liaoning provincial funding for scientific research projects (LJK0323)"

4. In this instance it seems there may be acceptable restrictions in place that prevent the public sharing of your minimal data. However, in line with our goal of ensuring long-term data availability to all interested researchers, PLOS’ Data Policy states that authors cannot be the sole named individuals responsible for ensuring data access (http://journals.plos.org/plosone/s/data-availability#loc-acceptable-data-sharing-methods).

Reviewers' comments:

Reviewer's Responses to Questions

**Comments to the Author**

1. Is the manuscript technically sound, and do the data support the conclusions?

Reviewer #1: Yes

Reviewer #2: Yes

Reviewer #3: Partly

2. Has the statistical analysis been performed appropriately and rigorously? 

Reviewer #1: Yes

Reviewer #2: Yes

Reviewer #3: N/A

3. Have the authors made all data underlying the findings in their manuscript fully available?

Reviewer #1: Yes

Reviewer #2: Yes

Reviewer #3: No

4. Is the manuscript presented in an intelligible fashion and written in standard English?

Reviewer #1: Yes

Reviewer #2: Yes

Reviewer #3: Yes

5. Review Comments to the Author

Reviewer #1: The article investigates the airflow and dust transport characteristics of parallel conveyor belt systems in coal preparation plants under different induced airflow conditions, focusing on dust The spray dedusting system is designed and implemented according to the pollution law of air flow, and the movement law of droplets after the spray system is arranged in the workshop is simulated and analyzed. The field test results show that the implementation of the dedusting system has achieved good results. The research content, methods, conclusions, and on-site applications of the article have good innovation and practical significance, and can provide a theoretical basis for dust pollution control in belt conveyor systems similar to coal preparation plants.

Reviewer #2: The subject of the study is the dust contamination mechanism under changing conditions of wind flow field in the workshop of a coal processing plant, which was investigated using a computational fluid dynamics based approach. The effects of induced airflow and corridor wind speed on the internal airflow field of the workshop were investigated, and the dust pollution mechanism under the changing conditions of the workshop wind field of the coal power plant was derived. The results of this study can provide an important reference for the study of dust pollution control in the workshop of coal processing plant. However, there are some minor problems in the article, which need to be slightly modified, and I support the publication of this paper.

1. In this paper, all variables in Equations 4, 5, 6, and 7 should be interpreted.

2. In Fig. 3, the curve represents the wind flow velocity, and the location of the measurement point of the curve should be labeled in the model plot.

3. On page 10 of the article, in Table 1, the boundary conditions are incomplete, please add them.

4. Throughout the article, the authors simulated the selection of 3 sets of corridor wind speeds as well as 3 sets of induced airflow values, please explain the reasons for the selections made.

5. Figures 5, 6, 7, and 8 are not formatted correctly, and the figure numbers should not be within the pictures.

6. This paper studies the coal preparation plant workshop spray dust reduction, but the authors for the selection of nozzles based on the selection and selection of models is not given, should be supplemented in the text.

7. Figure 9 is not clear enough, please ask the author to adjust and modify it.

8. Conclusion 2 of the article is not expressed with enough precision and summarized with enough fleshing out.

Reviewer #3: The article describe CFD simulations of dust distribution in a conveyor belt workshop. Mitigation measures using water sprays are also considered.

The introduction gives a good overview of the problem with dust, and of previous simulation studies of the problem. The computational methods should be better described, see comments below. The result section has a good description of the results, but the images need improvement. Overall, the study demonstrates a nice use of computational methods to improve working environment. I recommend publication of the study, considering the following comments are taken into account by the authors:

• The abstract should contain a short sentence with background on the problem.

• Equations for the turbulence model come a bit suddenly and out of context. Please first write the RANS Navier-Stokes equations and explain the connection to the turbulence model.

• Explain the boundary conditions better, please include a figure showing inlets, outlets, walls etc.

• What do you mean by cell mass values in the grid setup, and how does this relate to grid quality?

• What do you mean by “the airflow velocity in grid B already remains constant”, the velocity is clearly not constant?

• Fig 3, please have a better caption, the caption should explain the figure without needing to go into the text. The grid pictures should be bigger.

• In the text, you do not explain where this velocity in Fig 3 is extracted from?

• For the validation study, please give the exact coordinates of the measurement points. Also explain what equipment was used for the measurements.

• Fig 5-9, these should be bigger, they are very hard to interpret. Also, the captions should be more descriptive.

• The positions of the water mist devices should be clearly indicated.

6. PLOS authors have the option to publish the peer review history of their article (what does this mean?). If published, this will include your full peer review and any attached files.

Reviewer #1: No

Reviewer #2: No

Reviewer #3: No

---

## [Author Response · Author response to Decision Letter 0]

17 Jan 2024

Dear Editors and Reviewers:

Thank you for your letter and for the reviewers’ comments concerning our manuscript entitled “Research on factors affecting the spread of dust pollution in conveyor belt workshop and research on wet dust reduction technology” (ID: PONE-D-23-38475). All comments are very important and they are great support for our scientific research work. We have studied comments carefully and have made correction (Revised portion are marked in red in the text). The main corrections in the paper and the responds to the reviewer’s comments are as flowing:

Reviewer 1

The article investigates the airflow and dust transport characteristics of parallel conveyor belt systems in coal preparation plants under different induced airflow conditions, focusing on dust The spray dedusting system is designed and implemented according to the pollution law of air flow, and the movement law of droplets after the spray system is arranged in the workshop is simulated and analyzed. The field test results show that the implementation of the dedusting system has achieved good results. The research content, methods, conclusions, and on-site applications of the article have good innovation and practical significance, and can provide a theoretical basis for dust pollution control in belt conveyor systems similar to coal preparation plants.

Answers:Thank you for your professional evaluation of my manuscript, for your work I am deeply grateful, I will seriously listen to your suggestions to improve the article, I wish you a happy life!

Reviewer 2

The subject of the study is the dust contamination mechanism under changing conditions of wind flow field in the workshop of a coal processing plant, which was investigated using a computational fluid dynamics based approach. The effects of induced airflow and corridor wind speed on the internal airflow field of the workshop were investigated, and the dust pollution mechanism under the changing conditions of the workshop wind field of the coal power plant was derived. The results of this study can provide an important reference for the study of dust pollution control in the workshop of coal processing plant. However, there are some minor problems in the article, which need to be slightly modified, and I support the publication of this paper.

Answers: Thank you very much for the excellent and professional revisions of our manuscript. We have studied the reviewer's comments carefully and have made revisions that have been marked in red in the revised manuscript.

Comment 1)In this paper, all variables in Equations 4, 5, 6, and 7 should be interpreted.

Answers: Thank you for your comments.I have changed the variables to the correct format .

Comment 2)In Fig. 3, the curve represents the wind flow velocity, and the location of the measurement point of the curve should be labeled in the model plot.

Answers: Thank you for your comments.I have modified Figure 3 to mark the location of the measurement points

Comment 3)On page 10 of the article, in Table 1, the boundary conditions are incomplete, please add them.

Answers: Thank you for your comments.I have added to the conditions

Comment 4)Throughout the article, the authors simulated the selection of 3 sets of corridor wind speeds as well as 3 sets of induced airflow values, please explain the reasons for the selections made.

Answers: Thank you for your comments.The authors chose the three sets of wind speeds mainly based on the maximum and minimum wind speeds obtained from the actual measurements in the field to be used as the simulation conditions, and the average values of the measured data as the study conditions.

Comment 5) Fig 5, 6, 7, and 8 are not formatted correctly, and the figure numbers should not be within the pictures.

Answers: Thank you for your comments.Figure has been revised as requested and labeled in the article

Comment 6) This paper studies the coal preparation plant workshop spray dust reduction, but the authors for the selection of nozzles based on the selection and selection of models is not given, should be supplemented in the text.

Answers: Thank you for your comments.The dust reduction system uses pneumatic atomized nozzles, which are more suitable for capturing fine dust.

Comment 7) Fig 9 is not clear enough, please ask the author to adjust and modify it.

Answers:Thank you for your comments.Fig 9 has been adjusted in accordance with your suggestions.

Comment 8) Conclusion 2 of the article is not expressed with enough precision and summarized with enough fleshing out.

Answers: Thank you for your comments. I have added and summarized the conclusions based on your suggestions, which are much appreciated.

Thank you very much for your suggestion!

Reviewer 3

The article describe CFD simulations of dust distribution in a conveyor belt workshop. Mitigation measures using water sprays are also considered.The introduction gives a good overview of the problem with dust, and of previous simulation studies of the problem. The computational methods should be better described, see comments below. The result section has a good description of the results, but the images need improvement. Overall, the study demonstrates a nice use of computational methods to improve working environment. I recommend publication of the study, considering the following comments are taken into account by the authors:

Answers:Thank you for your professional evaluation of my manuscript, I have seriously revised your suggestions on my manuscript, your suggestions make my manuscript better, thank you for your guidance on my work, I wish you a happy life!

Comment 1) The abstract should contain a short sentence with background on the problem.

Answers: Thank you for your comments.I have revised the abstract as you suggested.

Comment 2) Equations for the turbulence model come a bit suddenly and out of context. Please first write the RANS Navier-Stokes equations and explain the connection to the turbulence model.

Answers: Thank you for your comments. The mathematical modeling equations section has been supplemented and refined.The RANS Navier-Stokes equations have been added.

Comment 3) Explain the boundary conditions better, please include a figure showing inlets, outlets, walls etc.

Answers: Thank you for your comments.The conditions of the model's inlets, outlets, and walls are described in additional detail in Manuscript 3.1.

Comment 4) What do you mean by cell mass values in the grid setup, and how does this relate to grid quality?

Answers: Thank you for your comments. Grid cell quality between 0.4 and 1 indicates that the set grid quality is good and meets the calculation requirements

Comment 5) What do you mean by “the airflow velocity in grid B already remains constant”, the velocity is clearly not constant?

Answers: Thank you for your comments.Grid B's airflow velocity has reached stabilization is very little affected by the grid, the previous inaccurate description has been revised in the article, thank you for your guidance.

Comment 6) Fig 3, please have a better caption, the caption should explain the figure without needing to go into the text. The grid pictures should be bigger.

Answers: Thank you for your comments. Fig 3 has been modified

Comment 7) In the text, you do not explain where this velocity in Fig 3 is extracted from?

Answers: Thank you for your comments.The speed measurement points have been labeled in the Fig 3.

Comment 8) For the validation study, please give the exact coordinates of the measurement points. Also explain what equipment was used for the measurements.

Answers: Thank you for your comments. Additional description of manuscript paragraph 3.4, location of measurement points and method of measurement.

Comment 9) Fig 5-9, these should be bigger, they are very hard to interpret. Also, the captions should be more descriptive.

Answers: Thank you for your comments. Changes have been made to Fig 5-9.

Comment 10) The positions of the water mist devices should be clearly indicated.

Answers: Thank you for your comments.A diagram of the location of the water mist unit has been added to the manuscript.（Fig 10）

Thank you very much for your suggestion!

---

## [Decision Letter · Decision Letter 1]

25 Jan 2024

PONE-D-23-38475R1Research on factors affecting the spread of dust pollution in conveyor belt workshop and research on wet dust reduction technologyPLOS ONE

Dear Dr. dong,

Thank you for submitting your manuscript to PLOS ONE. After careful consideration, we feel that it has merit but does not fully meet PLOS ONE’s publication criteria as it currently stands. Therefore, we invite you to submit a revised version of the manuscript that addresses the points raised during the review process.

**ACADEMIC EDITOR: Minor revision.**

We look forward to receiving your revised manuscript.

Kind regards,

Worradorn Phairuang, Ph.D.

Academic Editor

PLOS ONE

Journal Requirements:

Reviewers' comments:

Reviewer's Responses to Questions

**Comments to the Author**

1. If the authors have adequately addressed your comments raised in a previous round of review and you feel that this manuscript is now acceptable for publication, you may indicate that here to bypass the “Comments to the Author” section, enter your conflict of interest statement in the “Confidential to Editor” section, and submit your "Accept" recommendation.

Reviewer #1: All comments have been addressed

Reviewer #3: (No Response)

2. Is the manuscript technically sound, and do the data support the conclusions?

Reviewer #1: Yes

Reviewer #3: Yes

3. Has the statistical analysis been performed appropriately and rigorously? 

Reviewer #1: Yes

Reviewer #3: N/A

4. Have the authors made all data underlying the findings in their manuscript fully available?

Reviewer #1: Yes

Reviewer #3: Yes

5. Is the manuscript presented in an intelligible fashion and written in standard English?

Reviewer #1: Yes

Reviewer #3: Yes

6. Review Comments to the Author

Reviewer #1: (No Response)

Reviewer #3: The authors have adressed most of my comments. However, equations 2, 3 and 4 still need further revision. Equation 2 is not the continuity equation, it is the momentum equation, and it's not written correctly. Equations 3 and 4 are not consistent, equation 3 has the transient term, while equation 4 does not. Also, the convection term is written differently. Please also separate vector variables from scalars, either with an arrow or bold font. These minor mistakes tend to reduce confidence in your work.

7. PLOS authors have the option to publish the peer review history of their article (what does this mean?). If published, this will include your full peer review and any attached files.

Reviewer #1: No

Reviewer #3: No

---

## [Author Response · Author response to Decision Letter 1]

2 Feb 2024

Thank you for your comments.I have changed the variables to the correct format .Your advice to my academic play a guiding role, for the formula of the understanding I still have deficiencies, but after my study and ask for advice has been in accordance with your requirements for modification, thank you for your teaching!

---

## [Editor Report · Decision Letter 2]

9 Feb 2024

Research on factors affecting the spread of dust pollution in conveyor belt workshop and research on wet dust reduction technology

PONE-D-23-38475R2

Dear Dr. dong,

We’re pleased to inform you that your manuscript has been judged scientifically suitable for publication and will be formally accepted for publication once it meets all outstanding technical requirements.

Kind regards,

Worradorn Phairuang, Ph.D.

Academic Editor

PLOS ONE
---

## [Editor Report · Acceptance letter]

13 Feb 2024

PONE-D-23-38475R2 

PLOS ONE

Dear Dr. Dong, 

I'm pleased to inform you that your manuscript has been deemed suitable for publication in PLOS ONE. Congratulations! Your manuscript is now being handed over to our production team.

Kind regards, 

on behalf of

Assistant Professor Worradorn Phairuang 

Academic Editor

PLOS ONE